# OLGA: ONE-CLASS GRAPH AUTOENCODER

## ABSTRACT

One-class learning (OCL) comprises a set of techniques applied when real-world problems have a single class of interest. The usual procedure for OCL is learning a hypersphere that comprises instances of this class and, ideally, repels unseen instances from any other classes. Besides, several OCL algorithms for graphs have been proposed since graph representation learning has succeeded in various fields. These methods may use a two-step strategy, initially representing the graph and, in a second step, classifying its nodes. On the other hand, end-to-end methods learn the node representations while classifying the nodes in one learning process. We highlight three main gaps in the literature on OCL for graphs: (i) non-customized representations for OCL; (ii) the lack of constraints on hypersphere parameters learning; and (iii) the methods' lack of interpretability and visualization. We propose **O**ne-c**L**ass **G**raph **A**utoencoder (OLGA). OLGA is end-to-end and learns the representations for the graph nodes while encapsulating the interest instances by combining two loss functions. We propose a new hypersphere loss function to encapsulate the interest instances. OLGA combines this new hypersphere loss with the graph autoencoder reconstruction loss to improve model learning. OLGA achieved state-of-the-art results and outperformed six other methods with a statistically significant difference from five methods. Moreover, OLGA learns low-dimensional representations maintaining the classification performance with an interpretable model representation learning and results.

## 1 INTRODUCTION

Graphs offer a powerful structure for modeling real-world problems by explicitly capturing node relations. Edges represent their connections, providing a comprehensive view of interconnectedness and dependencies. This explicit representation of relations is essential in domains such as social networks, recommendation systems, and biological networks, enabling accurate analysis and prediction (Zhou et al., 2020; Xia et al., 2021; Deng et al., 2022).

Different classification problems involving an interest class are commonly modeled using graphs. These include tasks such as classifying fake news (de Souza et al., 2022), identifying events of interest (Nguyen & Grishman, 2018), predicting hit songs (da Silva et al., 2022), detecting fraud (Dou et al., 2020), and performing anomaly detection (Liu et al., 2022a). Notably, in many of these tasks, humans can naturally detect the interest class solely by observing instances of interest. In machine learning, this strategy is represented by One-Class Learning (OCL) approaches, which enable the classification of interest instances by training solely on samples from this class (Emmert-Streib & Dehmer, 2022). OCL reduces the need for extensive labeling efforts, is suitable for unbalanced scenarios, and does not require comprehensive coverage of the non-interest class(es) (Tax, 2001; Khan & Madden, 2014; Fernández et al., 2018; Alam et al., 2020).

OCL and graphs are attractive strategies for solving real-world problems with an interest class (Wang et al., 2021; Feng et al., 2022; Deldar et al., 2022). First, Two-step methods employed unsupervised graph neural networks (GNNs) to generate representations for graph nodes, followed by OCL algorithms to classify the interest nodes (Gôlo et al., 2022; da Silva et al., 2022; Huang et al., 2022b; Ganz et al., 2023). Recent advancements have focused on end-to-end one-class graph neural networks (OCGNNs), which simultaneously learn representations and classify instances as belonging to the interest class (Feng et al., 2022; Wang et al., 2021; Huang et al., 2021; Deldar et al., 2022). A common strategy in both approaches is defining a hypersphere to encapsulate the interest representations, aiming to separate them from non-interest (Tax & Duin, 2004; Wang et al., 2021).

Significant gaps need to be addressed when OCL and GNNs are combined. Firstly, two-step methods rely on pre-learning data representations, which are not customized for the OCL step (Wang et al., 2021). On the other hand, current end-to-end methods lack studies on graphs from different domains and lack constraints on hypersphere parameters learning, often leading to solutions trapped in local minima (Ruff et al., 2018; Feng et al., 2022; Wang et al., 2021). For instance, encapsulating the interest class around a single point in the learned space with a minimum-radius hypersphere can yield erroneous results for unseen data.

Another significant limitation in current research is the representation learning process interpretability issue, especially in OCL (Wu & Mooney, 2019; Liu et al., 2022b). Existing methods often assume high-dimensional latent spaces, which can hamper interpretability. In contrast, we argue that OCGNNs with low dimensions (two or three) offer natural interpretability. We can gain insights into the model's behavior by visually exploring the distance between instances in each learning epoch and visualizing the decision surface. Some studies have attempted interpretability by employing t-SNE for low-dimensional projections at the end of the learning process (Wang et al., 2021; Feng et al., 2022). However, we argue that explicitly learning low-dimensional node representations within the OCL context can benefit model interpretability and the overall OCL learning process.

This paper introduces OLGA (**O**ne-c**L**ass **G**raph **A**utoencoder), a new end-to-end one-class graph neural network method. OLGA achieves meaningful node representations and state-of-the-art one-class node classification by combining two loss functions. The first is based on the graph autoencoder loss, which aims to reconstruct and preserve the graph topology by mapping nodes into a new latent space. The second is a newly proposed hypersphere loss that leverages instances from the interest class to enhance one-class-oriented representation learning and classification. Moreover, OLGA also learns low-dimensional representations to improve the classification performance and introduce interpretability to the learning process. In summary, our contributions are:

1. We present OLGA, a new end-to-end one-class graph neural network method for node classification that simultaneously learns meaningful representations and classifies nodes.

2. We propose a new hypersphere loss function for one-class graph neural networks.

3. We demonstrate that low-dimensional one-class representations in OLGA support interpretability for model representation learning and improve classification performance.

4. We evaluate OLGA against other state-of-the-art methods on eight datasets from various domains and sources to confirm its performance in practical application scenarios.

We carried out an experimental evaluation using eight one-class datasets from diverse domains and sources. We compared OLGA with six other methods, including three OCGNNs with different GNN architectures and three strong baselines: Deep-walk, Node2Vec, and Graph Autoencoder (GAE) combined with the One-Class Support Vector Machine. OLGA achieved state-of-the-art performance on most datasets considering textual, image, and tabular domains outperforming the other six methods. Our method demonstrated a statistically significant improvement over five of the compared methods. Moreover, OLGA demonstrated the potential to learn low-dimensional representations enabling an interpretable learning process and data visualizations for OCL.

## 2 RELATED WORK

We divide the related work into two categories. The first category comprises the two-step methods. These methods generate embeddings through unsupervised graph neural networks and, in another step, apply a one-class learning algorithm. The second category comprises the end-to-end methods that learn the representations while classifying the nodes in one step (learning process).

### 2.1 TWO-STEP METHODS

Two-step methods apply graph neural networks (GNNs) in the first step. GNNs are state-of-the-art for representation learning on graphs. GNNs also capture structural features of the graphs and aggregate information from neighboring nodes associated with the interest class to generate embeddings for the nodes. Finally, in the second step, these methods use OCL algorithms to solve the one-class problem (da Silva et al., 2022; Huang et al., 2022b; Ganz et al., 2023; Gôlo et al., 2022).

In this sense, Gôlo et al. (2022) proposed to perform the movie recommendations through one-class learning. The authors used unsupervised graph neural networks with a link prediction loss to learn representations for the movies and users in the graph. The study used the One-Class Support Vector Machines (OCSVM) to classify.

da Silva et al. (2022) propose to detect hit songs through OCL. The authors used an unsupervised heterogeneous graph neural network to learn representations for the songs and, in the second step, used the OCSVM to classify and measure how famous a song is. Furthermore, Huang et al. (2022b) proposed to detect intrusions in systems through OCL. The authors proposed the One-Class Directed Heterogeneous graph neural network. This method combines an unsupervised heterogeneous GNN with the Deep Support Vector Data Description (DeppSVDD) algorithm (Ruff et al., 2018) to detect the anomalies. Ganz et al. (2023) detects backdoor software through OCL. Through collaborative graphs with commit nodes, branches, files, developers, and methods (functions), the authors represent the graph nodes through an unsupervised heterogeneous GNN, specifically a variational graph autoencoder (Kipf & Welling, 2016). In the second step, Ganz et al. (2023) uses DeppSVDD to detect backdoor software.

Two-step methods obtained state-of-the-art results in learning robust representations through GNNs for the OCL. However, these methods generate non-customized and agnostic representations for the OCL classification algorithm since the representation is learned independently of the OCL algorithm. This fact can limit the learned representation quality and negatively impact the OCL algorithm. End-to-end methods overcome this limitation by combining these two steps into one.

## 2.2 END-TO-END METHODS

End-to-end methods learn representations and classify instances in a single learning process. In the OCL, this is challenging due to the lack of counterexamples. In this sense, we need an appropriate loss function based on one class. In this way, the methods learn embeddings customized and non-agnostic that capture structural information from the graph, aggregating information from neighboring nodes and learning the pattern of the interest class while classifying instances (Feng et al., 2022; Wang et al., 2021; Huang et al., 2021; Deldar et al., 2022). In this sense, Feng et al. (2022) proposed to detect anomalies in IIoT Systems through OCL. The authors proposed using GNNs through a graph autoencoder to learn representations, and for classification, the authors used a threshold applied to the losses generated by the graph autoencoder.

In a pioneering way, Wang et al. (2021) proposed the one-class graph neural network (OCGNN), a graph neural network with a loss function similar to SVDD, learning embeddings for the nodes while encapsulating the interest instances through a hypersphere. The OCGNN was applied in graph anomaly detection (GAD). Later, Huang et al. (2021) proposed the GAD through the one-class temporal graph attention network method, a method similar to OCGNN that considers temporal information, attention mechanisms, and dynamic graphs with de hypersphere loss. Deldar et al. (2022) performs Android malware detection through a graph autoencoder (GAE) with a customized loss via thresholds. The authors used a loss function through an anomaly score generated by the difference between the initial representation and the representation decoded by the GAE.

Most end-to-end methods are based on the One-Class Graph Neural Network (OCGNN) from Wang et al. (2021). Formally, the OCGNN from Wang et al. (2021) minimizes Equation 1:

$$\mathcal{L}(r, \boldsymbol{W}) = \frac{1}{\nu |\boldsymbol{V}^{\text{in}}|} \sum_{i=1}^{|\boldsymbol{V}^{\text{in}}|} [\|g(\boldsymbol{V}, \boldsymbol{A}; \boldsymbol{W})_i - \boldsymbol{c}\|^2 - r^2]^+ + r^2 + \frac{\lambda}{2} \sum_{l=0}^{L} \|\boldsymbol{W}^l\|^2, \quad (1)$$

in which $\boldsymbol{V}$ is the node set, $\boldsymbol{V}^{\text{in}}$ is the interest node set, $\boldsymbol{A}$ is the adjacency matrix, $g(\boldsymbol{V}, \boldsymbol{A}; \boldsymbol{W})$ is a traditional graph neural network, $r$ is the radius of the hypersphere that is learned alternately with the neural network weights $\boldsymbol{W}$, $\nu \in [0, 1)$ is an upper bound on the fraction of training errors and a lower bound of the fraction of support vectors, $\boldsymbol{c}$ is the center of the hypersphere defined as the mean of the embeddings of the nodes in the class of interest through an initial forward propagation, $\lambda$ is the weight decay regularizer of the OCGNN, and $L$ is the number of neural network layers. This approach can be used with different GNNs, such as Graph Convolutional Networks

(GCN) (Kipf & Welling, 2017), Graph Attention Networks (GAT) (Velickovic et al., 2017), and GraphSAGE (Hamilton et al., 2017), by modifying the output of the term $g(\boldsymbol{V}, \boldsymbol{A}; \boldsymbol{W})$.

These end-to-end methods are applied in GAD, thus lacking studies on different domains. These methods based on hyperspheres lack constraints on the hypersphere loss function, often biasing the learning and harming the performance. For instance, in the OCGNNs, by using only the hypersphere loss function, all instances (interest and non-interest) will gradually converge to the center as GNN aggregates neighbor representations. Furthermore, in the GAEs with thresholds that use only the reconstruct loss, all instances (interest and non-interest) can converge for one single region since autoencoders are vulnerable to converge to a constant mapping onto the mean, which is the optimal constant solution of the mean squared error (Ruff et al., 2018). Another gap is the lack of methods exploited in low dimensionality to interpret and explain the learning process and to be used as visualization methods. Finally, GAEs are explorers in different studies (Deldar et al., 2022; Feng et al., 2022), and the hyperspheres loss function (Wang et al., 2021; Huang et al., 2021). However, GAEs were not exploited in OCGNNs that use the hypersphere loss function. One reason is the challenge of combining the loss function of the GAEs and hyperspheres. In this sense, in the next section, we present OLGA, the One-cLass Graph Autoencoder, a method to mitigate these gaps.

## 3 OLGA: ONE-cLASS GRAPH AUTOENCODER

We propose a novel end-to-end method for classifying interest nodes called One-cLass Graph Autoencoder (OLGA). OLGA learns node representations while classifying the nodes using hypersphere-based modeling (Tax & Duin, 2004). We base our method on a graph autoencoder to capture the structural properties of the graph through a reconstruction loss function (Kipf & Welling, 2016). Additionally, we propose a new loss function to encapsulate the interest instances that encourage these interest instances to approach the center even within the hypersphere. This characteristic encourages unlabeled interest instances to go into the hypersphere since labeled interest instances will remain updated even within the hypersphere. Otherwise, the learning process stabilizes, harming learning because unlabeled interest instance representations may not be updated.

We utilize the GAE architecture with our loss functions to improve the one-class learning in a multi-task learning way (Zhang & Yang, 2021; Sami et al., 2022). Multi-task learning aims to improve the learning for each task by leveraging the relevant information contained in multiple solved tasks (Zhang & Yang, 2018). In the context of OCL, multi-task learning has shown promise in improving the learning process (Xue & Beauseroy, 2016; Liu et al., 2021). Furthermore, promising multi-task GNNs indicate that performing multi-task learning also boosts each individual task while improving representation learning (Ma & Mei, 2019; Xie et al., 2020). In this context, we propose a multi-task learning approach using our loss functions for OCL in GNNs. Figure 1 illustrates OLGA.

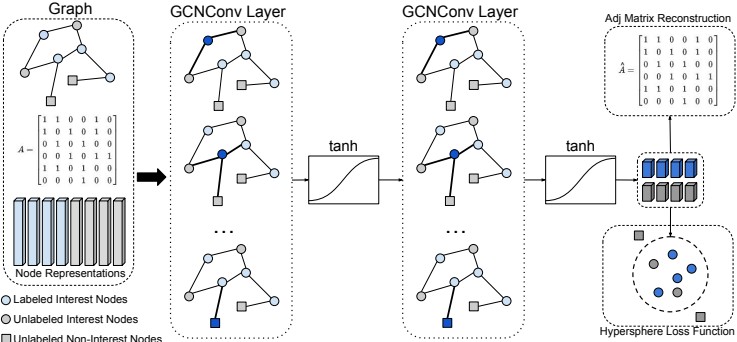

Figure 1: OLGA illustration with our new hypersphere and GAE loss functions.

Formally, a GAE utilizes a GNN encoder and a decoder represented by an inner product of the latent representation (Kipf & Welling, 2016). Equation 2 describes a GAE (Kipf & Welling, 2016).

$$GAE = \begin{cases} Encoder : \boldsymbol{H}^{(L)} = g(\boldsymbol{V}, \boldsymbol{A}; \boldsymbol{W}) \\ Decoder : \hat{\boldsymbol{A}} = \sigma(\boldsymbol{H}^{(L)} \cdot \boldsymbol{H}^{(L)\intercal}) \end{cases}, \tag{2}$$

in which, GAE learns $\boldsymbol{H}^{(L)}$ with the mean squared error between $\boldsymbol{A}$ and $\hat{\boldsymbol{A}}$, and $\sigma(.)$ is a logistic sigmoid function.

Our main task is the classification of instances as belonging to the interest class or not, which we define as $\mathcal{T}_1$. We define two additional reconstruction tasks to improve the learning and, consequently, the classification. The first one regards the reconstruction of the entire graph, defined as $\mathcal{T}_2$. The other task is the reconstruction of unlabeled nodes, defined as $\mathcal{T}_3$.

Let's define some sets of nodes according to $\boldsymbol{V}$. $\{\boldsymbol{V}^{\text{in}}, \boldsymbol{V}^{\text{u}}\} \in \boldsymbol{V}$, where $\boldsymbol{V}^{\text{in}}$ is the set of nodes of interest, and $\boldsymbol{V}^{\text{u}}$ is the set of unlabeled nodes. Let's define $d_i$ as the value indicating whether the interest instance is within the hypersphere with radius $r$ and center $\boldsymbol{c}$ given by Equation 3:

$$d_i = \|\boldsymbol{h}_i^{(L)} - \boldsymbol{c}\|^2 - r^2, \tag{3}$$

in which, $\boldsymbol{h}_i^{(L)} \in \boldsymbol{H}^{(L)}$. Negative values of $d_i$ indicate that the node is inside the hypersphere, and positive values indicate that it is outside.

For each task addressed in this study, we define a loss function. For $\mathcal{T}_1$, we propose a new loss function based on the hypersphere paradigm (Tax & Duin, 2004; Wang et al., 2021). The proposed hypersphere loss function penalizes instances of interest outside the hypersphere. Also, when the instance is inside the hypersphere, it continues to be penalized but to a lesser extent to encourage it to move closer to the center. We define our loss function in Equation 4. Figure $\mathcal{L}_1$ illustrates the loss function $\mathcal{L}_1$ in a one-dimensional space.

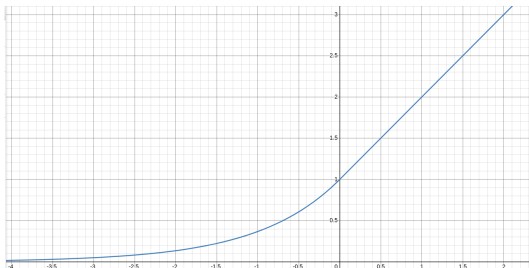

Figure 2: Our new hypersphere loss function ($\mathcal{L}_1$) Illustration.

$$\mathcal{L}_1(\boldsymbol{W}) = \frac{1}{|\boldsymbol{V}^{\text{in}}|} \sum_{i=1}^{\boldsymbol{V}^{\text{in}}} f(d_i), \tag{4} \qquad f(d_i) = \begin{cases} d_i + 1, & \text{if } d_i > 0 \\ \exp(d_i), & \text{otherwise} \end{cases}. \tag{5}$$

For $\mathcal{T}_2$, we define $\mathcal{L}_2$ as the reconstruction error of the entire graph ($\boldsymbol{V}^{\text{in}} \cup \mathcal{V}^u$), defined in Equation 6. For $\mathcal{T}_3$, we define the loss function $\mathcal{L}_3$, which can be defined as shown in Equation 7.

$$\mathcal{L}_2(\boldsymbol{W}) = mse(\boldsymbol{A}, \hat{\boldsymbol{A}}), \tag{6} \qquad \mathcal{L}_3(\boldsymbol{W}) = mse(\boldsymbol{A}^{\text{u}}, \hat{\boldsymbol{A}}^{\text{u}}), \tag{7}$$

in which $\boldsymbol{A}^{\text{u}}$ is the adjacency matrix of the unlabeled nodes in the graph, and $\hat{\boldsymbol{A}}^{\text{u}}$ is the reconstruction of this matrix generated by OLGA.

If we only use the $\mathcal{L}_1$, all instances will converge towards the center, regardless of whether they are interest nodes, as the GNN aggregates representations at each iteration. Therefore, we propose multi-task learning with additional loss functions to assist our main task solved by $\mathcal{L}_1$, combining the loss functions. Equation 8 presents our final loss. Our strategy combines the GAE loss function with the hypersphere loss function to work as a constraint so that the $\mathcal{L}_1$ does not entirely bias the learning and improve the learning of representations and classification performance.

$$\mathcal{L}(\boldsymbol{W}) = \mathcal{L}_1 * \alpha + \mathcal{L}_2 * \beta + \mathcal{L}_3 * \delta, \tag{8}$$

in which $\alpha$, $\beta$, and $\delta$ are the weights for $\mathcal{L}_1$, $\mathcal{L}_2$, and $\mathcal{L}_3$ losses in the training process.

The update of the network parameters is done based on the partial derivative of the losses. In the end, we have an addition of the partial derivatives for the $\frac{\partial \mathcal{L}_1}{\partial \theta_e} + \frac{\partial \mathcal{L}_2}{\partial \theta_d} + \frac{\partial \mathcal{L}_3}{\partial \theta_d}$, in which $\theta_e$ represents

the encoder parameters, i.e., $\boldsymbol{W}$ and bias, and $\theta_d$ the decoder, i.e., $\boldsymbol{W}$, bias, and the inner product. Since the $\mathcal{L}_2$ and $\mathcal{L}_3$ are an MSE based well defined in the literature, we focus on defining the $\mathcal{L}_1$ partial derivative (Equation 9). So that way, in Equation 10, we have a total partial derivative of

$$\frac{\partial \mathcal{L}_1}{\partial \theta_e} = \frac{1}{|\boldsymbol{V}^{\text{in}}|} \sum_{i=1}^{|\boldsymbol{V}^{\text{in}}|} \frac{\partial}{\theta_e} f(d_i), \qquad (9) \qquad \frac{\partial L_{\text{total}}}{\partial \theta} = \frac{\partial \mathcal{L}_1}{\partial \theta_e} + \frac{\partial \mathcal{L}_2}{\partial \theta_d} + \frac{\partial \mathcal{L}_3}{\partial \theta_d}. \qquad (10)$$

However, when combining two different loss functions, one potential issue is the lack of regularization in their scales, where one loss could have a stronger influence on the network's learning, causing the other task to be ignored entirely. However, using hyperbolic tangents serves as a solution to this problem. Note that in the adjacency matrix represented by zeros and ones, both $\mathcal{L}_2$ and $\mathcal{L}_3$ return values on a similar scale to those found in the matrix. In this sense, we need to focus on the one-class loss function to solve the problem. The hyperbolic tangent restricts the problem space to $-1$ and $1$ across all its dimensions. Therefore, the distance used for calculating $\mathcal{L}_1$ is on a similar scale as $\mathcal{L}_2$ and $\mathcal{L}_3$, avoiding the issue of imbalanced losses.

Once each dimension of the learned representation ranges between $-1$ and $1$ due to the hyperbolic tangent, the hypersphere's radius will be in the range $(0, 1]$. In this scenario, the hypersphere volume tends to $0$ the larger the dimension explored (Smith & Vamanamurthy, 1989). Therefore, with high dimensions, OLGA will learn representations for interest instances very close to the center until OLGA learns representations at a single point to encapsulate the interest instances. Even if the task is being solved in OCL, using these representations in a subtask becomes unfeasible.

If OLGA explores low dimensionality in its last layer (2 or 3), the volume of the hypersphere will not tend to $0$, which makes it possible for OLGA to learn representations that can be used in other tasks and still solve the node classification. In addition, when using low dimensions, OLGA becomes an interpretable representation learning method since we can visualize each learning stage by plotting the representations at each epoch. This fact also makes OLGA a method more interpretive. In addition, we can visualize the circle or sphere in the classification step.

## 4 EXPERIMENTAL EVALUATION

This section presents the experimental evaluation of this article. We present the used datasets, experimental settings, results, and discussion. Our goal is to demonstrate that our OLGA proposal outperforms other state-of-the-art methods. Another goal is to demonstrate that our method learns low-dimensional representations without losing classification performance and performing better than other methods. The experimental evaluation codes are publicly available[1].

### 4.1 DATASETS

We used one-class datasets from the literature from different domains, sources, and data types and used the $k$-nn modeling given the lack of benchmarking homogeneous graph datasets for one-class learning. We explored textual, image, and tabular datasets. We accurately choose one-class original datasets, i.e., datasets that have an interest class. We explore two tabular datasets collected from the UCI repository. The first is a dataset of musk and non-musk molecules (Chapman & Jain, 1994; Dietterich et al., 1993). Our interest class was musk molecules. The second is a malware detection dataset (Tuandromd) (Borah et al., 2020). Our interest class was malware in the Tuandromd dataset.

We also use three textual datasets. The first is a dataset about fake news detection (de Souza et al., 2022; Gôlo et al., 2021a). This dataset has real and fake classes, and our interest class was fake news. The dataset is called Fact Checked News (FCN) and is publicly available. The second is a dataset for detecting interest events (Gôlo et al., 2021b). This dataset has terrorism and non-terrorism classes, and our interest class was terrorism events. This dataset is also publicly available. The third textual dataset is for detecting relevant software reviews (Stanik et al., 2019; Gôlo et al., 2022). Our interest class was relevant reviews, while irrelevant reviews were considered outliers. The dataset is called App Reviews in English (ARE) and is not publicly available. However, the dataset creators (Stanik et al., 2019) make the datasets available for academic purposes upon request.

---

[1]**Ommited by double-blind review.**

In addition, we explore three image datasets. The first dataset was collected from the Kaggle platform (Antonov, 2019). This dataset is a dataset with food images and non-food images. We use food images as our interest class. We also collected from Kaggle the second dataset with images of lungs with and without pneumonia (Kermany et al., 2018). We used pneumonia images as an interest class. Finally, we used a dataset with images of healthy and abnormal strawberries (Choi et al., 2022). Healthy strawberries are our interest class.

We model the datasets using the similarity between nearest neighbors. In the graph, each node is connected with its $k$ nearest neighbors in this modeling. We use the values of $k = \{1, 2, 3\}$. The modeling that generated the best result in the validation set was used in the test set. The graph nodes must be represented to calculate the similarity between nodes. Tabular datasets have a natural and initial representation. However, text and image datasets are naturally unstructured.

For the text datasets, we use the pre-trained model Bidirectional Encoder Representations from Transformers (BERT) in the Multilingual version (Reimers & Gurevych, 2019) to represent the textual nodes since the use of BERT to initialize the textual representations nodes in GNNs presented good performances (Huang et al., 2022a). We used the Contrastive Language Image Pretraining (Radford et al., 2021) pre-trained model for the image datasets to represent the image nodes. In addition to using the initial representations to model our graphs, we also use these initial representations as input to methods based on OCGNNs, the unsupervised GAE, and our OLGA approach.

## 4.2 EXPERIMENTAL SETTINGS

We used three two-step baselines. In the first step, we generate the representations through an unsupervised graph-based method. In the second, we use the OCL algorithm to classify the instances based on the generated representations. As a representation method, we used DeepWalk (Perozzi et al., 2014), Node2Vec (Grover & Leskovec, 2016), and Graph Autoencoder (GAE) (Kipf & Welling, 2016). We used the One-Class Support Vector Machines (OCSVM) (Schölkopf et al., 2001) to classify the representations. We also compare our method with three state-of-the-art end-to-end algorithms based on the OCGNN proposed by (Wang et al., 2021). We use OCGCN, OCGAT, and OCSAGE, three OCGNN variations that consider the GCN (Kipf & Welling, 2017), the GAT (Velickovic et al., 2017), and the GraphSAGE (Hamilton et al., 2017) as layers of the GNN.

We use the 10-fold cross-validation adapted for OCL. We divided only the interest instances into 10 folds in the procedure since we used only the interest class to train. We separate 1 fold for the test and 9 folds for the training. Furthermore, all folders are used once as a test set. We add 50% of non-interest instances to the test set. For the validation set of each iteration, we use 10% of the training set and the other 50% of the non-interest instances. We use the $f_1$-macro to compare all models, as the $f_1$-macro is not biased by imbalance which is natural in OCL.

## 4.3 RESULTS AND DISCUSSION

Tables 1 and 2 present the results for the methods in the eight datasets. Table 1 presents the compared methods with embeddings of dimensions 128 (high) and OLGA with 2 or 3 (low). Table 2 presents OLGA and the other methods with 2 and 3 dimensions (low). Bold values indicate the best results. Underline values indicate second-best results. Our OLGA approach outperforms the other methods. OLGA obtained higher values of $f_1$-macro in four out of eight datasets considering Table 1 and five out of eight for Table 2. The most competitive method with OLGA was DeepWalk, which obtained better results in two datasets in the high embeddings scenario and two for low dimensions. OCGAT and GAE outperform the other methods in the other three datasets for high and low dimensions. OCGCN, OCSAGE, and GAE obtained the worst results in most cases.

OLGA outperforms the other methods in the two tabular datasets in Table 2 and in the Musk dataset in Table 1. Furthermore, OLGA also outperforms the other methods in datasets with textual content, obtaining higher $f_1$-macro in the Relevant Reviews dataset in Table 1 and 2. OLGA was able to detect relevant reviews satisfactorily. In the dataset with texts on terrorism and non-terrorism, Deepwalk + OCSVM outperformed OLGA. In the dataset with texts on fake and real news, GAE + OCSVM outperformed OLGA. Still, in the image datasets, OLGA obtains higher $f_1$-macro in two out of three datasets. We detected images of food and pneumonia satisfactorily. In the dataset with strawberry images, Deepwalk + OCSVM outperformed OLGA.

Table 1: 10-fold average for $f_1$-macro in the test set for **high dimension** results. Each line represents a dataset, and each column a method.

| Datasets | GAE | Deepwalk | Node2Vec | OCGCN | OCGAT | OCSAGE | OLGA |
|---|---|---|---|---|---|---|---|
| **Fakenews** | **0.942** | 0.879 | 0.829 | 0.746 | 0.884 | 0.824 | 0.940 |
| **Terrorism** | 0.727 | **0.981** | 0.800 | 0.845 | 0.900 | 0.785 | 0.921 |
| **Relevant R.** | 0.728 | 0.598 | 0.644 | 0.592 | 0.691 | 0.623 | **0.750** |
| **Food** | 0.982 | 0.752 | 0.960 | 0.901 | 0.994 | 0.985 | **0.997** |
| **Pneumonia** | 0.639 | 0.878 | 0.645 | 0.505 | 0.752 | 0.655 | **0.914** |
| **Strawberry** | 0.516 | **0.961** | 0.584 | 0.435 | 0.629 | 0.635 | 0.647 |
| **Musk** | 0.623 | 0.774 | 0.640 | 0.666 | 0.620 | 0.603 | **0.785** |
| **Tuandromd** | 0.904 | 0.689 | 0.905 | 0.898 | **0.978** | 0.974 | 0.974 |

Table 2: 10-fold average for $f_1$-macro in the test set for **low dimension** results. Each line represents a dataset, and each column a method.

| Datasets | GAE | Deepwalk | Node2Vec | OCGCN | OCGAT | OCSAGE | OLGA |
|---|---|---|---|---|---|---|---|
| **Fakenews** | **0.950** | 0.868 | 0.892 | 0.635 | 0.506 | 0.486 | 0.940 |
| **Terrorism** | 0.624 | **0.978** | 0.671 | 0.824 | 0.744 | 0.640 | 0.921 |
| **Relevant R.** | 0.742 | 0.614 | 0.707 | 0.546 | 0.541 | 0.521 | **0.750** |
| **food** | 0.995 | 0.768 | 0.980 | 0.787 | 0.635 | 0.526 | **0.997** |
| **Pneumonia** | 0.394 | 0.878 | 0.787 | 0.588 | 0.664 | 0.661 | **0.914** |
| **Strawberry** | 0.531 | **0.942** | 0.544 | 0.594 | 0.598 | 0.557 | 0.647 |
| **Musk** | 0.478 | 0.726 | 0.526 | 0.505 | 0.437 | 0.477 | **0.785** |
| **Tuandromd** | 0.863 | 0.689 | 0.823 | 0.756 | 0.598 | 0.854 | **0.974** |

In the two datasets in which we were competitive with Deepwalk + OCSVM, we observed a not natural imbalance, i.e., we have more interest instances in the test than non-interest (see Table 3). This imbalance is not natural in the real world since the interest class is usually a small sample of a larger universe. Also, one difference between OLGA and OCSVM is that OCSVM is inductive while OLGA is transductive. Therefore, our transductive method was not robust enough for this unbalanced scenario, while the OCSVM was not harmed because it is inductive.

Table 1 presents OLGA's performance considering low dimensions compared with the results from other methods with high dimensions. OLGA obtains better performances in four datasets (food, musk, pneumonia, and relevant reviews) with the advantage of being interpretable, explainable, and with the power of visualization. OLGA has competitive results in the other four datasets since OLGA obtains the second-best results when other methods outperform OLGA. We observe in table 2 that when another method obtains the highest $f_1$, OLGA obtains the second best result, i.e., compared to methods with high or low dimensionality, OLGA is competitive.

We performed Friedman's statistical test with Nemenyi's post-test (Trawinski et al., 2012) to compare the methods considering the low and high scenarios. Figure 3 presents a critical difference diagram generated through the Friedman test with Nemenyi's post-test result. The diagram presents the methods' average rankings. Methods connected by a line do not present statistically significant differences between them with 95% of confidence. The OLGA has the best average ranking with a statistically significant difference from all methods except Deepwalk. The Deepwalk obtained the second-best average ranking without statistically significant differences from the other methods. OCGCN and OCSAGE obtained the worst average rankings.

We chose a dataset of each data type, textual, image, and tabular, to present the representations generated by the OLGA learning. In this sense, Figure 4 presents the OLGA representations in the experimental evaluation on the relevant review, food, and tuandromd datasets. We present OLGA's learning process, i.e., four learning epochs at different stages. The first stage is epoch 0. The second is epoch 150. We chose 150 because the patience for this analysis was 300. The fourth stage is the epoch when the model converges, and the third stage is the average epoch between the second and fourth stages. Blue points represent the interest class, and green points the non-interest class.

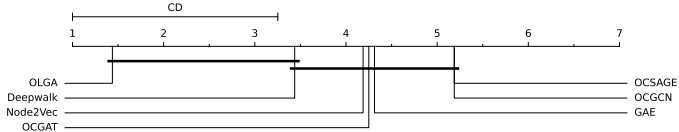

Figure 3: Critical difference diagram of Friedman's statistical test with Nemenyi post-test considering $f_1$-macro for low and high dimensional scenarios.

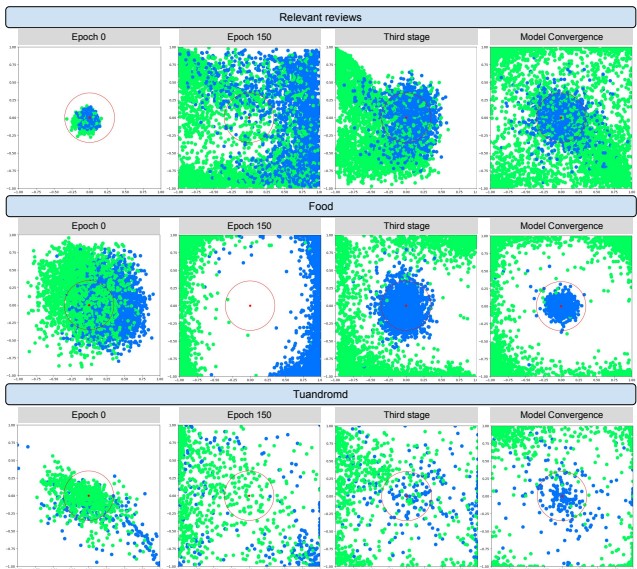

Figure 4: Two-dimensional representations of OLGA last layer consider the learned representations in three datasets. The colors indicate the interest class (blue) and the non-interest class (green).

We observe the entire learning process through representations of our method, shown in Figure 4. During stages one and two, we observe the method ignoring the hypersphere and focusing only on graph reconstruction (loss $\mathcal{L}_2$), which really is the OLGA goal. In the third stage, we observe multitask learning since the interest instances get closer to the hypersphere while non-interest instances are outside the hypersphere (losses $\mathcal{L}_1$ and $\mathcal{L}_3$). Finally, in the fourth stage, we observed that the one-class loss $\mathcal{L}_1$ encouraged the instances to continue coming to the hypersphere center, as proposed.

OLGA learns non-agnostic and customized representations for OCL, as shown in Figure 4. OLGA obtains the best performances, as shown above, with a representation with less overlap between classes, a one-class visualization more promising, and an interpretable representation learning model. We emphasize that combining OLGA loss functions in a multi-task way and using low dimensions allowed our proposal to be interpretable, explicable, and used for visualization.

## 5 CONCLUSIONS AND FUTURE WORK

We propose OLGA, an end-to-end graph neural network for OCL. OLGA combines a hypersphere and reconstruction loss functions. We also introduce a novel hypersphere loss function that encapsulates the interest instances and encourages these instances to approach the center even within the hypersphere. The learning process of OLGA allows us to explore low-dimensional representations during the classification process without harming the classification performance, providing interpretability and visualization. Our approach outperforms other state-of-the-art methods, showing statistically significant differences from five out of six compared methods, and generates visually meaningful representations suitable for OCL.

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

## A    APPENDIX

### A.1    EXPERIMENTAL EVALUATION DETAILS

We use the following parameters for the methods:

- **DeepWalk and Node2Vec:** window size = 10, embedding size = $\{2, 3, 128\}$, walks per node = 15, walk length = 10, q=1 and p=4;
- **GAE:** epochs=100, learning rate=0.01, optimizer=$Adam$, embedding size = $\{2, 3, 128\}$;
- **OCSVM:** degree for polynomial kernel =$\{2\}$, $\gamma = \{scale, auto\}$, $\nu = \{0.05 * \iota, 0.005 * \iota; \iota \in [1..17]\}$, and $kernels = \{rbf, linear, sigmoid, polynomial\}$;
- **OCGCN, OCGAT, OCSAGE and OLGA:** epoch = 5000, patience = $\{300, 500\}$, learning rate= $\{0.0001, 0.0005\}$, and optimizer=$Adam$;
- **OCGCN, OCGAT, and OCSAGE:** $\nu = \{0.001, 0.01, 0.1, 0.2, 0.3, 0.4, 0.5\}$, weight decay = 0.0005, dropout = 0.5, and activation = {ReLU}, embedding size = $\{2, 3, 128\}$;
- **OLGA:** embedding size = $\{2, 3\}$, center = 0, radius = $\{0.3, 0.35, 0.4\}$, and activation = {TanH}. Until the patience/2 epoch, we use $\alpha = 0$, $\beta = 1$, and $\delta = 0$. After the patience/2 epoch, we use $\alpha = 1$, $\beta = 0$, and $\delta = 1$.

Table 3 presents the details of the datasets such as dataset type, number of interest instances (#I), number of non-interest instances (#NI), number of interest instances for training (#Tr), number of interest instances for the test (#TeI), and number of non-interest instances for the test (#TeNI). We obtain these values according to the 10-fold cross-validation.

Table 3: dataset properties.

| Datasets | Type | #I | #NI | #Tr | #TeI | #TeNI |
|----------|------|------|------|------|------|-------|
| **FakeNews** | Text | 1044 | 1020 | 940 | 104 | 510 |
| **Terrorism** | Text | 5926 | 721 | 5333 | 593 | 360 |
| **Relevant R.** | Text | 2537 | 3855 | 2283 | 254 | 1927 |
| **Food** | Image | 2500 | 2500 | 2250 | 250 | 1025 |
| **Pneumonia** | Image | 4273 | 1583 | 3846 | 427 | 791 |
| **Strawberry** | Image | 3367 | 153 | 3030 | 337 | 76 |
| **Musk** | Tabular | 1224 | 5850 | 1102 | 122 | 2925 |
| **Tuandromd** | Tabular | 3565 | 899 | 3209 | 356 | 449 |

## A.2 DIMENSION ANALYSIS FOR OUR HYPERSPHERE LOSS FUNCTION

We show the hypersphere volume for the OLGA loss function considering the values of the radius used (0.3, 0.45, 0.5). We show the volume considering dimensions from 1 to 40. Figure 5 shows the analysis result. The green line represents the 0.5 radius, the orange represents the 0.45, and the blue represents the 0.3. The axis y represents the volume, and the axis x represents the dimension. We use a red line to indicate the volume obtained with our dimensions numbers. We note that with dimensions higher than 15, the volume tends to be 0 regardless of radius. With smaller dimensions, we observe that the volume does not tend to be 0, allowing OLGA to better represent learning.

The volume was generated by:

$$V_n(r) = \frac{\pi^{n/2}}{\Gamma(\frac{n}{2} + 1)} r^n,$$ (11)

in which, $r$ is the radius, $n$ is number of dimension and $\Gamma$ is the Euler's gamma function.

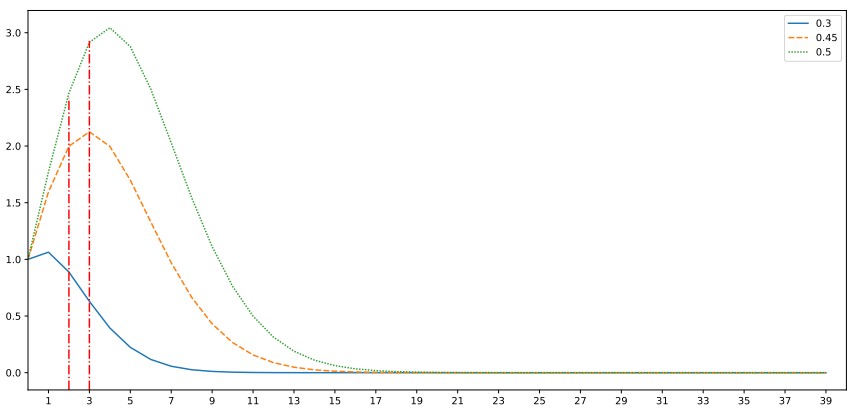

Figure 5: Relations between the hypersphere volume and the dimension of the representation used.

## A.3 RESULTS WITH STANDARD DEVIATION AND COMPLETE 2D PLOT

Table 4: Averages and standard deviation for $f_1$-macro in the test set for **high dimension** results. Each line represents a dataset, and each column a method. Bold values indicate the best results.

| Datasets | GAE | Deepwalk | Node2Vec | OCGCN | OCGAT | OCSAGE | OLGA |
|---|---|---|---|---|---|---|---|
| **Fakenews** | **0.942 (0.011)** | 0.879 (0.016) | 0.829 (0.022) | 0.746 (0.045) | 0.884 (0.017) | 0.824 (0.015 | 0.940 (0.016) |
| **Terrorism** | 0.727 (0.038) | **0.981(0.003)** | 0.800 (0.015) | 0.845 (0.021) | 0.900 (0.007) | 0.785 (0.020) | 0.921 (0.007) |
| **Relevant R.** | 0.728 (0.009) | 0.598 (0.023) | 0.644 (0.009) | 0.592 (0.010) | 0.691 (0.006) | 0.623 (0.010) | **0.750 (0.007)** |
| **Food** | 0.982 (0.004) | 0.752 (0.012) | 0.960 (0.007) | 0.901 (0.009) | 0.994 (0.003) | 0.985 (0.003) | **0.997 (0.002)** |
| **Pneumonia** | 0.639 (0.012) | 0.878 (0.010) | 0.645 (0.012) | 0.505 (0.012) | 0.752 (0.011) | 0.655 (0.014) | **0.914 (0.010)** |
| **Strawberry** | 0.516 (0.021) | **0.961 (0.007)** | 0.584 (0.031) | 0.435 (0.093) | 0.629 (0.020) | 0.635 (0.025) | 0.647 (0.101) |
| **Musk** | 0.623 (0.063) | 0.774 (0.006) | 0.640 (0.019) | 0.666 (0.018) | 0.620 (0.008) | 0.603 (0.013) | **0.785 (0.029)** |
| **Tuandromd** | 0.904 (0.011) | 0.689 (0.007) | 0.905 (0.007) | 0.898 (0.008) | **0.978 (0.004)** | 0.974 (0.004) | 0.974 (0.006) |

Table 5: Averages and standard deviation for $f_1$-macro in the test set for **low dimension** results. Each line represents a dataset, and each column a method. Bold values indicate the best results.

| Datasets | GAE | Deepwalk | Node2Vec | OCGCN | OCGAT | OCSAGE | OLGA |
|---|---|---|---|---|---|---|---|
| **Fakenews** | **0.950 (0.010)** | 0.868 (0.018) | 0.892 (0.015) | 0.635 (0.014) | 0.506 (0.013) | 0.486 (0.022) | 0.940 (0.016) |
| **Terrorism** | 0.624 (0.011) | **0.978 (0.004)** | 0.671 (0.014) | 0.824 (0.012) | 0.744 (0.012) | 0.640 (0.010) | 0.921 (0.007) |
| **Relevant R.** | 0.742 (0.009) | 0.614 (0.017) | 0.707 (0.008) | 0.546 (0.007) | 0.541 (0.005) | 0.521 (0.008) | **0.750 (0.007)** |
| **food** | 0.995 (0.003) | 0.768 (0.012) | 0.980 (0.003) | 0.787 (0.016) | 0.635 (0.015) | 0.526 (0.012) | **0.997 (0.002)** |
| **Pneumonia** | 0.394 (0.000) | 0.878 (0.009) | 0.787 (0.010) | 0.588 (0.010) | 0.664 (0.013) | 0.661 (0.006) | **0.914 (0.010)** |
| **Strawberry** | 0.531 (0.017) | **0.942 (0.013)** | 0.544 (0.022) | 0.594 (0.023) | 0.598 (0.015) | 0.557 (0.013) | 0.647 (0.101) |
| **Musk** | 0.478 (0.068) | 0.726 (0.015) | 0.526 (0.013) | 0.505 (0.008) | 0.437 (0.007) | 0.477 (0.011) | **0.785 (0.029)** |
| **Tuandromd** | 0.863 (0.020) | 0.689 (0.009) | 0.823 (0.012) | 0.756 (0.011) | 0.598 (0.016) | 0.854 (0.012) | **0.974 (0.006)** |

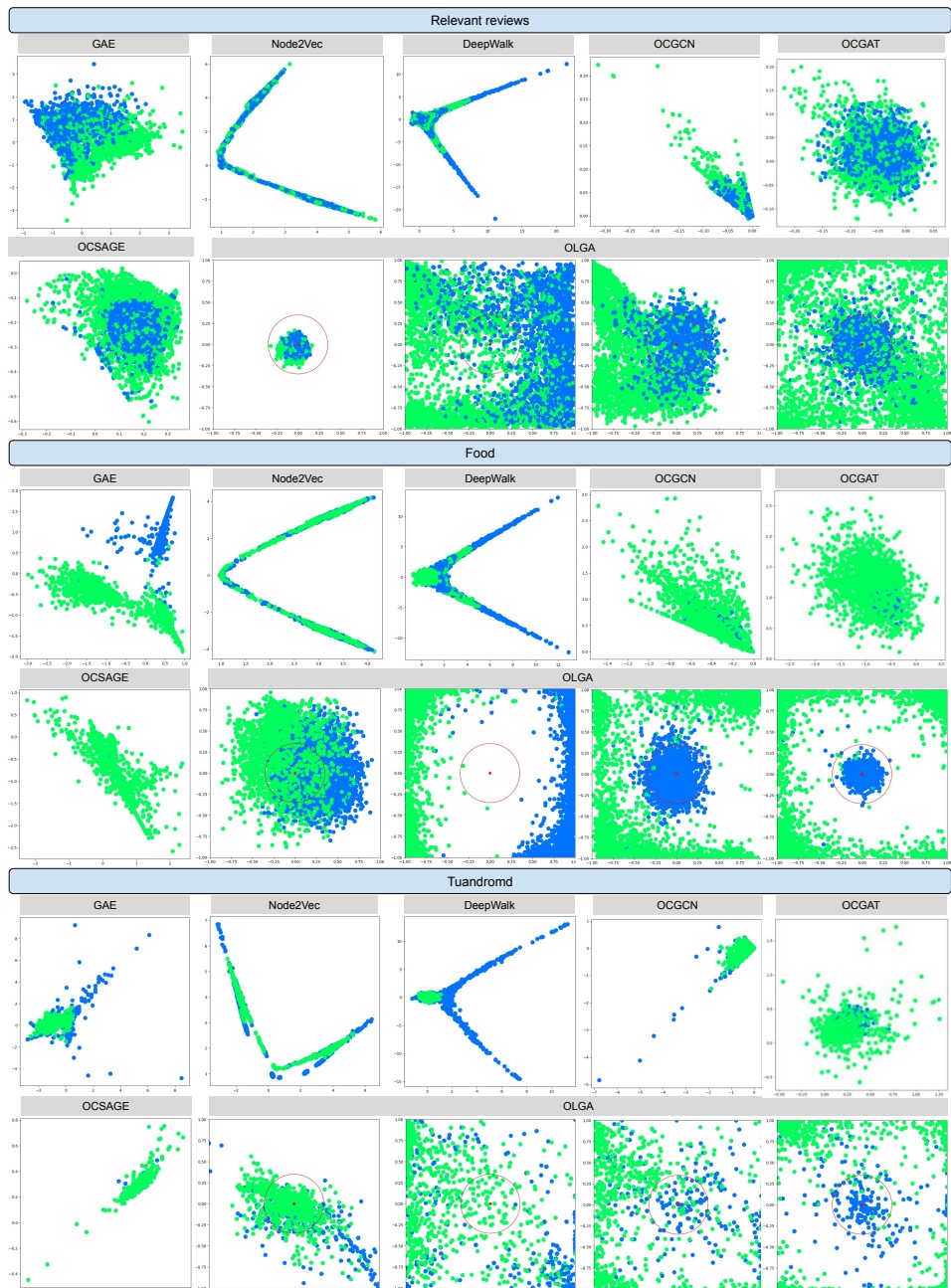

Figure 6: Two-dimensional representations of each method consider the learned representations in three datasets. The colors indicate the interest class (blue) and the non-interest class (green). Representation methods that show less overlap between classes and a one-class visualization are more promising.

Figure 6 shows that the two-step methods generated non-customized representations for one-class learning. Even GAE generates some good representations, which partially separate the interest and non-interest classes in most cases, these representations are not customized because they are agnostic to the classification model. End-to-end methods based on OCGNN generate representations non-agnostic to the classification model. However, OCGNN does not customize the representations enough for a hypersphere to classify interest and non-interest instances. OLGA also learns non-agnostic representations such as OCGNNs. On the other hand, OLGA generates customized representations for one-class learning, obtaining the best performances, as shown above.

