# OpenReview forum: "OLGA: One-cLass Graph Autoencoder"
_ICLR.cc/2024/Conference — Submitted to ICLR 2024_

### Official Review · Reviewer_iTUK · 2023-10-23

**Soundness:** 1 poor
**Presentation:** 1 poor
**Contribution:** 1 poor
**Rating:** 3
**Confidence:** 4

**Summary:**

This paper proposed that a reconstruction-based loss function can improve one-class learning models in determining whether an instance belongs to the target class or not. The proposed method introduced in this study is termed "One-Class Graph Autoencoder" (OLGA).

**Strengths:**

1. When combining different loss functions,   this paper considers the scale regularization.
2. The paper conducts a comprehensive review of the relevant literature.

**Weaknesses:**

1. Many claims in the article have not been reasonably explained.

     a. The rationale behind the benefits of low-dimensional node representations within the OCL context for enhancing model interpretability is not adequately elucidated.

     b. The reason behinds the integration of the autoencoder framework into one-class classification models is not well illustrated. In page 5, authors wrote "If we only use the L1, all instances will converge towards the center, regardless of whether they are interest nodes, as the GNN aggregates representations at each iteration. Therefore, we propose multi-task learning with additional loss functions to assist our main task solved by L1, combining the loss functions." Such explanation is too high-level, and the underlying mechanism why autoencoder is beneficial to one-class classification is still unclear.

      c. The paper presents the challenge of combining GAEs and hypersphere loss functions, which it appears to resolve by simply combining these functions without a detailed explanation of the challenges involved.

      d. The paper attempts to address the hypersphere collapse problem in one-class classification using low-dimensional representations, but this paper does not sufficiently discuss the advantages of lower dimensions compared with existing solutions. Furthermore, the explanation of why lower dimensions enhance interpretability remains unclear.

2. The writing of the article is not professional enough. In introduction, author wrote "encapsulating the interest class around a single point in the learned space with a minimum-radius hypersphere can yield erroneous results for unseen data".   However,  this paper does not respond it in the introduction section, even in the paragraph discussing the proposed method OLGA.
3. The choice of comparison algorithms in the paper is not sufficiently novel. For example, the OCSAGE method, which uses GraphSAGE, was introduced in 2018, and deep one-class classification was also proposed in the same year. This lack of innovative comparison algorithms diminishes the paper's contribution.
4. In the reference literature, there are too few articles from conferences such as ICLR, NeurIPS, and ICML. A more comprehensive reference selection from these esteemed conferences would enhance the paper's credibility.

Overall, the current version does not meet the acceptance criteria of ICLR.

**Questions:**

Please refer to the weakness section.

---

> ### Author Response · Authors · 2023-11-17
>
> 1. Thank you for pointing out this issue. We support the issue of interpretability in relation to the dimension with more facts and references. Basically, without the help of other methods, such as t-SNE, OCL methods for graphs naturally do not offer interpretability.
>
> "Interpretability has gained increasing attention, driven by the demand for model transparency (Carvalho et al., 2019; Gilpin et al., 2018). Generally, interpretability can be categorized into distinct types, including feature-based, model-based, and latent space-based methods (Das et al., 2020; Guidotti et al., 2018; Gilpin et al., 2018). With the rise of representation learning models, there has been a growing interest in interpretability centered around the latent space (Gilpin et al., 2018; Chen et al., 2016). Latent space-based methods delve into the internal representations learned by models. Many studies explore t-SNE (Van der Maaten & Hinton, 2008) for embedding visualization (Wang et al.,2021) to visualize these representations in lower dimensions and provide interpretability. While these methods are invaluable, they are not inherently integrated into the representation learning process. However, methods that inherently yield low-dimensional representations while preserving classification performance enable mapping these representations to two or three dimensions, offering users an intuitive means to interact with and comprehend the model's decisions."
>
> See these references in the PXa1 reviewer response.
>
> 2. Thanks for pointing out this gap. The main motivation for using the GAE reconstruction loss function is to create a constraint so that the hypersphere loss function does not completely bias learning since this total bias can leave all instances within the hypersphere. We added more information about this in the introduction.
>
> “(...) The first is based on the graph autoencoder loss, which aims to reconstruct and preserve the graph topology by mapping nodes into a new latent space. The second is a newly proposed hypersphere loss that leverages instances from the interest class to enhance one-class-oriented representation learning and classification. The GAE reconstruction loss function works as a constraint so that the hypersphere loss function does not map all inputs inside the hypersphere. (...)”.
>
> 3. Thanks for pointing out this issue. The challenge is related to the scales of the loss functions. We modified the section where we highlighted the challenge to be clearer. Furthermore, at the end of our proposal section, we discuss the scales of the loss functions and what we did to maintain the same scales.
>
> “(...) However, GAEs were not exploited in OCGNNs that use the hypersphere loss function. One reason is the challenge of combining the loss function of the GAEs and hyperspheres since we have different scales for the loss functions. In this sense, in the next section, we present OLGA, the One-cLass Graph Autoencoder, a method to mitigate these gaps.”
>
> “However, when combining two different loss functions, one potential issue is their scales, where one loss could have a stronger influence on the network's learning, causing the other task to be ignored entirely. However, using hyperbolic tangents serves as a solution to this problem. (...)”
>
> 4. We show in the appendices that using our proposal with high dimensionality generates the collapse of the hypersphere (all points converge to 0). Furthermore, when using low dimensions, we can provide interpretability (now, this is clearer in the introduction) since the representations can be seen at each learning epoch. We agree that the work needs more foundation for this part of interpretability. Therefore, we provide more information and explanation with a new paragraph on why low-dimensional representations provide more interpretability, as presented in item 1.
>
> Finally, in 2021, when Wang proposed OCGNN, it obtained state-of-the-art results for one-class learning on graphs. Therefore, we mainly compared this method and not with others proposed before 2021 that obtained worse results than OCGNN. We highlight a scarcity of end-to-end one-class learning methods proposed after 2021 with reproducible code.

---

> > ### Comment · Reviewer_iTUK · 2023-11-20
> > **Thanks for the response**
> >
> > After carefully reviewing all the responses from the authors, I have chosen to maintain the current score.

---

### Official Review · Reviewer_Yigw · 2023-10-25

**Soundness:** 2 fair
**Presentation:** 3 good
**Contribution:** 3 good
**Rating:** 5
**Confidence:** 4

**Summary:**

In this work, the authors introduce OLGA, an end-to-end One-class Graph Autoencoder that combines a novel hypersphere loss function with a graph autoencoder reconstruction loss to encapsulate interest instances and achieve state-of-the-art results, outperforming several methods.

**Strengths:**

- One-class learning is a very fundamental problem for graph-related problems, and exploring one-class learning on graphs is a very interesting topic.
- The paper is well-organized and easy to be understood.

**Weaknesses:**

- Adding a diagram in the introduction to illustrate the significance and importance of one-class learning on graphs would be beneficial.
- The lack of innovation is a concern, as the author has primarily combined two conventional and commonly used loss functions.
- I suggest the author provide detailed statistics about the dataset.
- The authors could provide an anonymous GitHub link to ensure the reproducibility of the paper.
- Due to the popularity of large language models, it is advisable for the authors to consider using Large Language Models (LLMs) or other encoders to initialize text representations for comparison.

**Questions:**

See above.

---

> ### Author Response · Authors · 2023-11-17
>
> 1. Thanks for the suggestion. We added an illustration in the appendices because of limited space reasons.
>
> 2. We would like to point out that the proposed hypersphere loss function is new. The way of combining loss functions using the same scales is also new. To the best of our knowledge, no other work has proposed a hypersphere loss function encouraging examples to go to the center when they are inside the hypersphere, nor has it combined hypersphere and GAE loss functions, maintaining their scales.
>
> 3. The statistics were in the paper. We moved them to the appendix for space reasons.
>
> 4. The work has GitHub and will be released as soon as it is published. We avoid using links to avoid breaking the double-blind since some reverse engineering cases are being applied to anonymous GitHub to find the original creators.
>
> 5. We appreciate your suggestion and agree that we can make improvements by modifying the initial representation. Unfortunately, we do not have the time to run all the experiments again, but we will use this strategy in future work. However, the study focuses on the proposed method for one-class learning, agnostic to the data. Thus, seeking more robust initializations could improve the results and expand the work's scope.

---

> > ### Comment · Reviewer_Yigw · 2023-11-22
> >
> > The author has partially addressed my concerns and I will keep my score unchanged.

---

### Official Review · Reviewer_PXa1 · 2023-10-28

**Soundness:** 2 fair
**Presentation:** 2 fair
**Contribution:** 2 fair
**Rating:** 3
**Confidence:** 2

**Summary:**

In this paper, the authors concentrate on One-Class Learning (OCL) with the aim of addressing three specific limitations. They introduce an innovative one-class graph autoencoder named OLGA, which serves as an end-to-end graph learning framework. Additionally, the authors propose two distinct loss functions to encapsulate the interesting instances.

**Strengths:**

1.The author provides a statement about OCL.
2.The author identifies and discusses three issues in OCL.

**Weaknesses:**

1.In the introduction, the author claims, "Existing methods often assume high-dimensional latent spaces, which can hamper interpretability." However, this statement lacks a basis in terms of theoretical or experimental analysis. As a result, the subsequent experimental results do not provide evidence that high-dimensional features are significantly worse or even better than low-dimensional features, such as OCGAN, OCGAT, and OCSA. This raises doubts about the accuracy of the author's description.

2.There is a scarcity of comparative methods in the paper, and recent research is underrepresented. Out of the six comparative methods, three are derived from the same paper.

3.The third motivation point mentions "the methods' lack of interpretability and visualization." Nevertheless, the paper itself lacks corresponding theoretical explanations.

4.The paper lacks a complexity analysis of the proposed method, both theoretically and experimentally.

5.In Formula 8, there are three hyperparameters. Unfortunately, the paper does not include a sensitivity analysis for these parameters, making it difficult to intuitively gauge the effectiveness of the design module.

6.The innovation of the author's method appears limited, as it seems to be based on GAE and two MSE losses.

**Questions:**

See Weakness

---

> ### Author Response · Authors · 2023-11-17
>
> 1. Thanks for pointing out this issue. We removed this statement from the introduction as suggested.
>
> 2. In 2021, when Wang proposed OCGNN, it obtained state-of-the-art results for one-class learning on graphs. Therefore, we mainly compared this method and not with others proposed before 2021 that obtained worse results than OCGNN. We highlight a scarcity of end-to-end one-class learning methods proposed after 2021 with reproducible code.
>
> 3. Thank you for pointing out this issue. We support the issue of interpretability in relation to the dimension with more facts and references. Basically, without the help of other methods, such as t-SNE, OCL methods for graphs naturally do not offer interpretability.
>
> "Interpretability has gained increasing attention, driven by the demand for model transparency (Carvalho et al., 2019; Gilpin et al., 2018). Generally, interpretability can be categorized into distinct types, including feature-based, model-based, and latent space-based methods (Das et al., 2020; Guidotti et al., 2018; Gilpin et al., 2018). With the rise of representation learning models, there has been a growing interest in interpretability centered around the latent space (Gilpin et al., 2018; Chen et al., 2016). Latent space-based methods delve into the internal representations learned by models. Many studies explore t-SNE (Van der Maaten & Hinton, 2008) for embedding visualization (Wang et al.,2021) to visualize these representations in lower dimensions and provide interpretability. While these methods are invaluable, they are not inherently integrated into the representation learning process. However, methods that inherently yield low-dimensional representations while preserving classification performance enable mapping these representations to two or three dimensions, offering users an intuitive means to interact with and comprehend the model's decisions."
>
> Diogo V Carvalho, Eduardo M Pereira, and Jaime S Cardoso. Machine learning interpretability: A survey on methods and metrics. Electronics, 8(8):832, 2019.
>
> Leilani H Gilpin, David Bau, Ben Z Yuan, Ayesha Bajwa, Michael Specter, and Lalana Kagal. Explaining explanations: An overview of interpretability of machine learning. In 2018 IEEE 5th International Conference on data science and advanced analytics (DSAA), pp. 80–89. IEEE, 2018.
>
> Saikat Das, Namita Agarwal, Deepak Venugopal, Frederick T Sheldon, and Sajjan Shiva. Taxonomy and survey of interpretable machine learning method. In Proceedings of the Symposium Series on Computational Intelligence, pp. 670–677. IEEE, 2020.
>
> Riccardo Guidotti, Anna Monreale, Salvatore Ruggieri, Franco Turini, Fosca Giannotti, and Dino Pedreschi. A survey of methods for explaining black box models. ACM computing surveys (CSUR), 51(5):1–42, 2018.
>
> Xi Chen, Yan Duan, Rein Houthooft, John Schulman, Ilya Sutskever, and Pieter Abbeel. Infogan: Interpretable representation learning by information maximizing generative adversarial nets. Advances in neural information processing systems, 29, 2016.
>
> 4. We appreciate the comment, but due to time constraints, we did not carry out experiments comparing algorithm execution times. Furthermore, we emphasize that other OCL studies have already carried out this analysis, and we developed an algorithm with the same strategy, i.e., we don't have a significant difference in time compared to the OCGNN.
>
> 5. We decided to vary the other parameters and keep these three fixed. We chose these fixed values to activate and deactivate the loss functions during learning, as this characterizes the learning of OLGA. Therefore, we only use the configuration reported in the appendices of these three parameters, both in validation and testing. It is worth mentioning that this does not prevent other works from exploring intermediate values as extensions (values between 0 and 1).
>
> 6. The innovations are: (i) In the new hypersphere loss function ($\mathcal{L_1}$); (ii) The combination obeying the scales of our new hypersphere loss function and the GAE reconstruction function that resulted in the OLGA approach; and (iii) a new interpretable approach for one-class learning in graphs.

---

### Official Review · Reviewer_3sUJ · 2023-10-31

**Soundness:** 1 poor
**Presentation:** 2 fair
**Contribution:** 1 poor
**Rating:** 3
**Confidence:** 4

**Summary:**

This paper presents OLGA, a new method for one-class learning on graphs. OLGA uses a graph autoencoder and a novel hypersphere loss function to learn node representations and classify them as interest or non-interest. OLGA also learns low-dimensional representations that enable interpretability and visualization of the learning process and the data. The paper evaluates OLGA on eight datasets from various domains and sources, and shows that it outperforms six other methods.

**Strengths:**

1. The manuscript introduces a novel end-to-end method for one-class learning on graphs called OLGA, which combines a graph autoencoder and a hypersphere loss function.
2. The manuscript proposes a new hypersphere loss function that encourages the interest instances to approach the center of the hypersphere.
3. The manuscript evaluates OLGA on eight datasets from various domains and sources, and shows that it outperforms six other methods.

**Weaknesses:**

1. The motivation for using GAE is not well explained. Also, how simply introducing the graph autoencoder loss into one-class learning can help to solve the three gaps introduced in the abstract. A more detailed discussion should be added.
2. It is unclear how the function in Eq.5 is derived.
3. For the two reconstruction losses, A contains topology information of unlabeled nodes, setting a reconstruction loss on A^u could be repetitive and meaningless. The authors should give more explanation.
4. The writing should be improved. Many discussions are hard to understand.
5. A mathematical problem formulation would be helpful.

**Questions:**

NA

---

> ### Author Response · Authors · 2023-11-17
>
> 1. Thanks for pointing out this gap. The main motivation for using the GAE reconstruction loss function is to create a constraint so that the hypersphere loss function does not completely bias learning since this total bias can leave all instances within the hypersphere. We added more information about this in the introduction.
>
> “(...)  The first is based on the graph autoencoder loss, which aims to reconstruct and preserve the graph topology by mapping nodes into a new latent space. The second is a newly proposed hypersphere loss that leverages instances from the interest class to enhance one-class-oriented representation learning and classification. The GAE reconstruction loss function works as a constraint so that the hypersphere loss function does not map all inputs inside the hypersphere. (...)”.
>
> 2. We added the derivative of equation 5 to the paper.
>
> $f'(d_i) = 1, ~~ \text{if} ~~ d_i > 0 ~~~  e^{d_i}, ~~ \text{otherwise.} $
>
> 3. The idea of separating into two reconstruction loss functions is so that only the reconstruction of unlabeled nodes is used when using the hypersphere loss function. This is clear in the definition of the alpha, beta, and gamma parameters in the appendices, where L2 and L3 are not used together during learning. We agree that this may confuse the reader. Therefore, we change L2 to reconstruct only the labeled nodes of interest. Therefore, we will not have a repeat. Furthermore, we can use the new L2 and L3 together when reconstructing the entire graph.
>
> $\mathcal{L}_{2}(\mathbf{{W}}) = mse(\mathbf{A^{in}}, \hat{\mathbf{A}}^{\text{in}})$,
>
> $\mathcal{L}_{3}(\mathbf{{W}}) = mse(\mathbf{A^{u}}, \hat{\mathbf{A}}^{\text{u}})$,
>
> "in which $\mathbf{A}^{\text{u}}$ is the adjacency matrix of the unlabeled nodes in the graph, $\hat{\mathbf{A}}^{\text{u}}$ is the reconstruction of this matrix generated by OLGA, $\mathbf{A}^{\text{in}}$ is the adjacency matrix of the labeled nodes, and $\hat{\mathbf{A}}^{\text{in}}$ is the reconstruction of the $\mathbf{A}^{\text{in}}$."
>
> 4. We carried out an extensive reading of the paper to improve writing and discussions.
>
> 5. We insert a formulation of the problem.
>
> "We define OCL for graphs as a function $ocl: \mathcal{V} \rightarrow \mathbf{y}$ that maps a node $v_i \in \mathcal{V}$ to a value $y_i$ that indicates the confidence of a node $v_i$ belonging to the class of interest training only with positive labels of the interest node set $\mathcal{V}^{in}$. Thus, classification through OCL aims to learn a function $ocl^*$, which approximates the unknown mapping function $ocl$."

---

### Author Response · Authors · 2023-11-17

The authors would like to thank the reviewers for their precious time and valuable comments. We have carefully addressed all the comments. We provided a point-by-point to the reviewer’s comments. We present the corresponding changes and refinements made in the revised article below:

First, we add a new paragraph on the issue of interpretability in the introduction section.

"Interpretability has gained increasing attention, driven by the demand for model transparency (Carvalho et al., 2019; Gilpin et al., 2018). Generally, interpretability can be categorized into distinct types, including feature-based, model-based, and latent space-based methods (Das et al., 2020; Guidotti et al., 2018; Gilpin et al., 2018). With the rise of representation learning models, there has been a growing interest in interpretability centered around the latent space (Gilpin et al., 2018; Chen et al., 2016). Latent space-based methods delve into the internal representations learned by models. Many studies explore t-SNE (Van der Maaten & Hinton, 2008) for embedding visualization (Wang et al.,2021) to visualize these representations in lower dimensions and provide interpretability. While these methods are invaluable, they are not inherently integrated into the representation learning process. However, methods that inherently yield low-dimensional representations while preserving classification performance enable mapping these representations to two or three dimensions, offering users an intuitive means to interact with and comprehend the model’s decisions."

Second, we added extra information on the motivation for the GAE reconstruction function in the introduction section.

"(...) The first is based on the graph autoencoder loss, which aims to reconstruct and preserve the graph topology by mapping nodes into a new latent space. The second is a newly proposed hypersphere loss that leverages instances from the interest class to enhance one-class-oriented representation learning and classification. The GAE reconstruction loss function works as a constraint so that the hypersphere loss function does not map all inputs inside the hypersphere. (...)”.

Third, we insert a formulation of the problem in the related work section.

"We define OCL for graphs as a function $ocl: \mathcal{V} \rightarrow \mathbf{y}$ that maps a node $v_i \in \mathcal{V}$ to a value $y_i$ that indicates the confidence of a node $v_i$ belonging to the class of interest training only with positive labels of the interest node set $\mathcal{V}^{in}$. Thus, classification through OCL aims to learn a function $ocl^*$, which approximates the unknown mapping function $ocl$."

Fourth, we insert more details for combining the loss functions in the related work.

“(...) However, GAEs were not exploited in OCGNNs that use the hypersphere loss function. One reason is the challenge of combining the loss function of the GAEs and hyperspheres since we have different scales for the loss functions. In this sense, in the next section, we present OLGA, the One-cLass Graph Autoencoder, a method to mitigate these gaps.”

---

### Meta-Review · Area_Chair_eBdQ · 2023-12-06

**Metareview:**

This paper studies the problem of one-class learning on graphs and proposes a One Class Graph Autoencoder (OLGA) that combines a hypersphere loss function to capture the data/class of interest and a classic reconstruction loss. The experimental evaluation shows a promising behavior.

On the positive side, the reviews have highlighted that the paper studies an interesting problem with a good positioning/discussion with respect to the literature.
On the negative side, the contribution appears rather incremental, the writing has to be improved: many parts are not clear enough and need more explanations/justifications, the mathematical formalization needs to be improved, the experimental evaluation has also some limits.

Authors provided a rebuttal to answer the points raised by reviewers with a general remark on the interpretability issue and the interest of the reconstruction loss.
The answers of the authors have addresses some issues, but they did not change the general evaluation of the reviewers who globally think that the contribution is limited in its current form.

As a consequence, I propose rejection.

**Justification For Why Not Higher Score:**

3 reviewers evaluated the paper as a direct reject and the rebuttal did not change their general evaluation.

**Justification For Why Not Lower Score:**

N/A

---

### Decision · Program_Chairs · 2024-01-16

Reject